# Explore barriers to using the internet for health information access in African countries: A systematic review

**Alex Ayenew Chereka**[1]*, **Adamu Ambachew Shibabaw**[1], **Fikadu Wake Butta**[1], **Mathias Nega Tadesse**[2], **Mekashaw Tareke Abebe**[3], **Fekadu Ayelgn Atanie**[4], **Gemeda Wakgari Kitil**[5]

1 Department of Health Informatics, College of Health Sciences, Mattu University, Mattu, Ethiopia, 2 Department of Computer Science, College of Engineering and Technology, Kebri Dehar University, Kebri Dehar, Ethiopia, 3 Department of Accounting and Finance, College of Business and Economics, University of Gondar, Gondar, Ethiopia, 4 Department of Laboratory, College of Medicine and Health Sciences, Bahirdar University, Bahirdar, Ethiopia, 5 Department of Midwifery, College of Health Sciences, Mattu University, Mattu, Ethiopia

* zemeneayenew@gmail.com

## Abstract

### Background

The Internet is a crucial source of health information, providing access to vast volumes of high-quality, up-to-date, and relevant healthcare information. Its impact extends beyond information access, influencing medical practice through the widespread adoption of tele-medicine and evidence-based medicine. Despite the significant global increase in internet usage, Africa lags in internet penetration, particularly in utilizing the internet for health information. This study aims to systematically review the literature to explore barriers to accessing health information on the Internet in African countries.

### Methods

The study was conducted from January 1 to February 28, 2023. It followed the Preferred Reporting Items for Systematic Reviews and Meta-Analyses (PRISMA) guideline to systematically review published studies investigating the utilization of the Internet for health information in African countries. A comprehensive search was conducted across various databases, including Google Scholar, PubMed, Cochrane Library, Hinari, CINAHL, and Global Health. The inclusion criteria were applied, resulting in the selection of six studies that formed the basis for our analysis.

### Result

This systematic review identifies eleven barriers to accessing health information on the internet. These include a lack of ownership of smart electronic devices, infrequent internet use, limited internet access, low E-health and computer literacy, slow internet connection, high cost of internet access, insufficient information search skills, residing in rural areas, dealing with diverse fields, and having low perceptions.

**Data availability statement:** All the data used for the current study are available within the manuscript itself.

**Funding:** The author(s) received no specific funding for this work.

**Competing interests:** The authors have declared that no competing interests exist.

## Conclusion

Improving our understanding of barriers to accessing health information online is essential for policymakers, governments, academics, and healthcare professionals. To enhance the use of the Internet for health information and strengthen the overall health system, policymakers should prioritize increasing Internet accessibility, reducing costs, improving connections, offering basic computer skills training, and ensuring the availability of electronic devices in all institutions.

### Author summary

The internet is an essential source of health information, offering individuals the opportunity to access timely and accurate data to inform health decisions. However, in African countries, substantial barriers hinder the effective use of this resource. This systematic review investigates these barriers, identifying key challenges such as socio-economic inequalities, limited digital literacy, high costs of internet access, and inadequate infrastructure. Cultural factors, mistrust of online health information, and the lack of content tailored to local languages and contexts further exacerbate these issues. Our findings emphasize the need for targeted interventions to overcome these barriers. Strategies such as improving digital infrastructure, reducing internet costs, enhancing digital health literacy, and developing culturally relevant health content are critical for promoting equitable access to online health information. Addressing these barriers has the potential to improve health outcomes, reduce disparities, and foster digital inclusion in healthcare across African countries. This study provides valuable insights for researchers, policymakers, and stakeholders seeking to understand and address the digital health divide. It serves as a foundation for designing effective policies and programs to support internet-based health information access, ultimately contributing to better health equity in African settings.

## 1. Background

The Internet was created to help people quickly share information, and it's widely used for various purposes, including accessing health-related information [1–3]. Both healthcare providers and the general public find the Internet to be a valuable source of health information [4]. Since 1990, there has been a significant increase in using the Internet for health-related searches [5]. This trend provides healthcare professionals with easy access to a vast amount of high-quality, up-to-date, and relevant healthcare information [6]. People around the world are increasingly turning to the Internet to find health information and support [7,8].

The Internet, through forums, websites, blogs, and social networks, has become a prominent platform for promoting and preventing health issues among both professionals and the general public [9,10]. Moreover, the Internet has transformed medical practices by facilitating the widespread use of telemedicine and evidence-based medicine [11]. This shift has made it common for individuals worldwide to seek health information online, demonstrating the Internet's crucial role in improving the quality of healthcare information and making it more accessible [7,8].

According to reports from the Pew Internet and American Life Project in 2000 and 2009, 55% and 80% of adults in the United States who had Internet access used it to get clinical or health information [7,12,13]. This shows a significant increase in Internet use for

health-related purposes. However, earlier studies suggested that many people didn't have a strong intention to use the Internet for health reasons [14,15]. A more recent study in 2022 found that out of 5.4 billion Internet users worldwide, only 50 million of them use the Internet for health-related information. This number is quite low when compared to the total number of Internet users globally [16,17]. Despite the growth in Internet usage, only a small fraction of people are actively seeking health information online, indicating a potential gap between the availability of health resources online and people's inclination to use them [16,17].

Studies reveal varying rates of internet usage for health information across different countries, with the UK having a notably high rate of 97% of people using the internet for this purpose [18], Jeddah recorded 92.7% [19], Nigeria physicians 90% [20], Malaysia 80% [21], Ghana university students 67.7% [22], Kuwait 62.9 [23], united state 61% [24], Addis Ababa Ethiopia 59.1% [25], Spanish 55.7% [26], Ghana adolescents 53% [27], and Gondar Ethiopia 47.4% [28] were used internets for health information. Interestingly, these numbers suggest that the habit of using the Internet for health information is generally lower in developing countries compared to developed countries [29]. The disparities in internet use for health purposes across different regions highlight potential variations in access, awareness, or cultural factors that influence people's reliance on online platforms for health-related information [7,8]. Understanding these differences is crucial for developing targeted strategies to improve health information accessibility globally [16,17].

In Africa, internet penetration is below the global average, standing at 43% in 2021 compared to the global average of 66% [29]. This lower internet penetration in Africa contributes to a limited use of the internet for health information, especially when compared to the total number of internet users globally [16,17]. Despite the African health policy emphasizing essential health services for all, certain challenges hinder its realization [30]. These challenges include an insufficient number of healthcare professionals relative to the population, mismanagement of funds, and low awareness, willingness, attitude, perceived usefulness, infrastructure, professional deference, academic status, age category, and year of study among university or college students, among other factors [14,19,21,23,25,27,31–34]. These attributes play a crucial role in determining the utilization of the internet for health information.

Even though various studies have explored internet use for health information, there's a notable gap in a comprehensive systematic review specific to the African continent. This systematic review aims to pinpoint barriers to using the internet for health information-seeking in Africa. The findings are expected to shed light on the broader challenges related to internet use in the African context. This understanding can be pivotal for health managers, policymakers, and planners, enabling them to tailor interventions that specifically improve internet utilization for health information in African countries. Identifying key determinant factors is a crucial aspect of this process.

## 2. Methods

### 2.1. Source of information and search strategy

This systematic review and meta-analysis followed the Preferred Reporting Items for Systematic Reviews and Meta-Analyses (PRISMA) criteria [35]. The study's progression at each review level was illustrated using the PRISMA flow diagram. The PRISMA flow diagram outlines the number of records identified, included, and excluded, along with justifications for exclusions. A comprehensive search was conducted in MEDLINE (via PubMed), Google Scholar, Scopus, and Web of Science.

To systematically review and estimate the aggregated internet usage for health information and its associated factors in Africa, we employed a literature review approach in this

study. Information professionals conducted the literature search, identifying relevant sources through online databases, including Medline, PubMed, Scopus, Cochrane, EMBASE, African Journal Online (AJOL), HINARI, and Science Direct. Additionally, a search for grey literature was performed using Google, Google Scholar, and other internet search engines to identify any additional articles up to the end of February 2023.

To find all internet usage for health information in Africa, we employed various search phrases, including Medical Subject Headings (Mesh), keywords, and free text search queries. For the identification of publications in online databases, a systematic search was conducted using the following combination of search phrases: (("use" OR "advantage" OR "importance" AND "internet" AND "health information") AND ("associated factors" OR "determinants")). The search was specifically limited to studies in Africa that examined the extent of internet utilization for health information and its associated factors.

**Study period:** This systematic review was conducted from January 1 to February 30, 2023. The literature search, data extraction, and analysis were carried out during this period to explore the barriers to using the internet for health information access in African countries.

## 2.2. Inclusion and exclusion criteria

**2.2.1. Inclusion criteria of this study.** The inclusion criteria for this review encompassed articles primarily conducted in Africa. Eligible studies focused on internet usage for health information as their final variable, reporting its prevalence. Similarly, studies utilizing quantitative research techniques among healthcare professionals in all healthcare settings were included. This criterion extended to both peer-reviewed journal articles and grey literature published in English. Additionally, articles examining aspects related to internet utilization for health information, those published in scholarly journals or conference proceedings, and those using Africa as the country of data collection were included.

**2.2.2. Exclusion criteria of this study.** In the process of study selection, we applied specific exclusion criteria to ensure the precision and relevance of our investigation. Studies characterized by incomplete texts, challenges in data extraction, publications in languages other than English, indistinctly categorized outcome variables, and those reporting on the utilization of the internet for non-health information in African nations were deliberately omitted from our analysis. Furthermore, to maintain the focus on substantive research and scholarly contributions, papers lacking editorial reports, letters, reviews, or commentaries were purposefully excluded from consideration in this study. This meticulous approach to exclusion criteria was implemented to refine the scope and quality of the research included in our analysis.

## 2.3. Measurements of internet users for health information

The primary focus of this systematic review and meta-analysis was to ascertain the collective extent of individuals utilizing the Internet for health-related purposes across African countries. Specifically, the measurement of internet use for health information was based on a binary response, where participants were categorized as either "Yes" or "No" based on whether they employed the internet to access health-related information. This measurement allowed us to determine the proportions of individuals within the study population who actively sought health information through online resources.

## 2.4. Data extraction

During this phase, data extraction was conducted with a meticulous approach. Initially, one investigator meticulously compiled relevant information from the included studies into a

shared computer-based spreadsheet. Subsequently, a second investigator diligently reviewed the spreadsheet to ensure consistency and accuracy in the extracted data.

For each study, a comprehensive set of details was gathered, encompassing the initial author, publication year, number of participants, background information, study regions, sample size, data collection techniques, and the study's design. Additionally, the extraction process included obtaining data on the level of readiness to adopt electronic medical record systems and related metrics, all presented with 95% confidence intervals.

Furthermore, any identified issues or discrepancies were noted, and a thorough discussion with the corresponding author is planned to ensure clarity and accuracy in the extracted information. This rigorous data extraction process forms the foundation for the subsequent stages of our analysis and contributes to the overall robustness of the study.

### 2.5. Evaluation of the selected literature's quality

To ascertain the robustness of the selected literature, we conducted a meticulous evaluation of each study's quality. This assessment was carried out using a standardized tool designed to categorize potential bias and provide insights into any disparities in the findings among the included studies. Both authors meticulously scrutinized the methodological aspects and other key elements of each publication. For this purpose, we employed a modified version of the Newcastle-Ottawa Scale (NOS) tailored for cross-sectional research—an acknowledged and valid instrument for gauging bias risk in observational studies [36]. In our analysis, we deemed works scoring 7 or higher on the modified NOS components as particularly relevant, thus ensuring a rigorous selection process. To further enhance the reliability of our evaluation, three independent reviewers performed a meticulous quality control check, contributing to the overall credibility of our assessment.

### 2.6. Patient and public involvement

No patients or the public took part in this study.

## 3. Result

### 3.1. Study selections

A total of 510 articles were initially identified across multiple databases. Due to duplication, 121 kinds of literature were excluded. In addition, 113 articles were dropped through a review of titles and abstracts. Moreover, 262 full-text articles were excluded due to the study area and 8 articles were excluded due to the unavailability of the full text. Finally, 6 full-text articles were to be eligible for systematic review (Fig 1).

### 3.2. Characteristics of included studies

This study comprises six cross-sectional quantitative articles conducted in varied educational and healthcare settings. Two articles delved into university contexts, exploring various departments and fields of study [22,25], Furthermore, two articles were carried out in teaching hospitals, honing in on healthcare professionals and physicians [20,28], Another article focused on a college environment, encompassing all students [37], Lastly, one article specifically addressed adolescent girls in a secondary school setting [38]. This diversity in study settings and participant demographics, spanning different educational levels and healthcare backgrounds, enhances our overall comprehension of internet use for health information across a wide spectrum of demographic groups (Table 1).

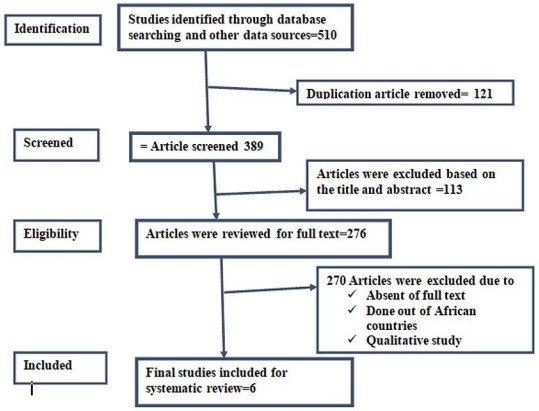

**Fig 1. PRISMA flowchart showing the selection process of the article.**

**Table 1. Characteristics of included articles.**

| Authors and references | Countries | Publication year | Study design | Prevalence | Sample size/sampling stratagem | Data collection technique |
|---|---|---|---|---|---|---|
| Osei Asibey B. [22] | Ghana | 2017 | Cross-sectional | 67.7 | 740 | Questionnaire |
| Ajuwon GA. [20] | Nigeria | 2006 | Cross-sectional | 90 | 172 | Questionnaire |
| Ibegbulam et al. [38] | Nigeria | 2018 | Cross-sectional | 96 | 120 | Questionnaire |
| Shiferaw KB, Mehari EA. [28] | Ethiopia | 2018 | Cross-sectional | 47.4 | 291 | Questionnaire |
| Derseh MH, et al. [25] | Ethiopia | 2022 | Cross-sectional | 59.1 | 845 | Questionnaire |
| Abukari Z. [37] | Ghana | 2021 | Cross-sectional | 96 | 161 | Questionnaire |

## 3.3. Barriers to using the internet for health information in Africa

The six studies revealed a total of 11 barriers, detailed in Table 2. These barriers encompassed issues such as insufficient e-health literacy, varied fields of study among students, limited computer literacy, lack of ownership of smart electronic devices, inadequate internet access, slow internet connection hindering health information retrieval, high costs associated with internet use for health information, limited information search skills, rural residences among healthcare professionals, and low perception of the internet's utility for health information. These identified barriers emerged as significant challenges impeding the effective utilization of the internet for health information purposes.

The 11 identified barriers were categorized based on their frequency of occurrence across the studies, with the most frequently mentioned ones listed first. The breakdown of occurrence rates for these barriers is as follows: "Lack of smart electronic device ownership" and "Low frequent use of the internet" were reported in five out of six studies (83.3%); "Lack of internet access" appeared in four out of six studies (66.6%); "Lack of E-health literacy," "Lack of computer literacy," "Slow internet connection," "Cost of internet access," "Lack of information search skill," and "Rural residence" each appeared in three out of six studies (50%); and "Variety Field of study among students" and "Low perception to use the internet for health information" each appeared in two out of six studies (33.3%). This frequency analysis provides insights into the prevalence and significance of each barrier across the examined studies.

**Table 2. Barriers to using the internet for health information in Africa and frequency occurrences.**

| No | Barriers | Reference | Frequencies | Percentage |
|---|---|---|---|---|
| 1 | Lack of E-health literacy | [25,28,38] | 3/6 | 50 |
| 2 | Variety of fields of study among students | [25,37] | 2/6 | 33.3 |
| 3 | Lack of smart electronic device ownership | [22,28,37,38] | 4/6 | 66.67 |
| 4 | Lack of Internet access | [20,28,37,38] | 4/6 | 66.67 |
| 5 | Lack of computer literacy | [22,37,38] | 3/6 | 50 |
| 6 | Slow internet connection | [22,37,38] | 3/6 | 50 |
| 7 | Cost of Internet access | [20,22,37,38] | 4/6 | 66.67 |
| 8 | Lack of information search skill | [20,22,37,38] | 4/6 | 66.67 |
| 9 | Rural residence | [25,28,37] | 3/6 | 50 |
| 10 | Low frequent use of the internet | [22,28,37,38] | 4/6 | 66.67 |
| 11 | Low perception of using the Internet for health information | [28,37] | 2/6 | 33.3 |

## 4. Discussion

The exploration of barriers to internet use for health information access in African countries is critical for understanding and addressing disparities in health information accessibility and, ultimately, health outcomes. Globally, the internet has become an indispensable resource for health information, supporting individuals in enhancing health literacy, making informed health decisions, and managing their health conditions independently [39,40].

However, in many African contexts, a combination of structural, socio-economic, and cultural obstacles significantly restricts internet access, potentially exacerbating health disparities and undermining the effectiveness of public health interventions in these regions [40,41].

This systematic review identifies several key barriers to internet use for health information access in African countries, underscoring the need for targeted policies and interventions to address these challenges. Notably, barriers such as lack of smart device ownership, limited internet access, high internet costs, insufficient information-searching skills, and infrequent internet use were reported in over 60% of the studies analyzed. Additionally, factors like slow internet connections, low e-health literacy, lack of computer literacy, and rural residence were observed in more than half of the studies. These barriers, individually and collectively, significantly hinder the ability to utilize the Internet for health information, impacting health outcomes and exacerbating disparities in resource-limited settings.

The absence of smart devices, such as smartphones, tablets, laptops, or computers, stands out as a critical barrier to accessing online health information. Without these devices, individuals are unable to engage with digital health resources, significantly limiting their ability to obtain health information via the Internet. Numerous studies have demonstrated that device ownership is a strong predictor of internet use, with those lacking access to such devices being far less likely to seek health information or engage with online health tools [42]. In areas where the digital divide is significant, the availability of a device is not simply a convenience but an essential requirement for accessing health information. The inability to afford or access these devices further exacerbates inequalities in health literacy and outcomes, particularly in regions like Africa, where socio-economic challenges heighten these barriers [37,38,43].

Similarly, Limited internet access and high connectivity costs are additional critical barriers. In many African countries, the infrastructure required for reliable internet access is either inadequate or expensive. The prohibitive cost of internet services, particularly in rural areas, limits the duration and frequency with which individuals can access the internet, entrenching the digital divide The financial burden of internet access is especially significant for

low-income populations, who are less able to afford regular online engagement [44]. Research has consistently shown that high internet costs restrict access to online health information and prevent individuals from fully utilizing digital health tools, contributing to disparities in health information access [20,38,41].

Moreover, insufficient information-searching skills contribute to low engagement with online health resources. While internet access may be available, individuals often lack the necessary skills to effectively search for, evaluate, and utilize health information. This limitation is particularly significant in communities where digital literacy remains low, and where the internet is predominantly used for social media or entertainment rather than educational or health-related purposes. The ability to critically assess online health information is crucial to ensure that users are accessing reliable, evidence-based content and making informed health decisions [41]. Without proper information-searching skills, even those who have access to the internet are less likely to benefit from the wealth of health resources available online [22,37].

Infrequent internet use compounds the challenges associated with accessing online health information. Even when individuals have the necessary devices and internet access, irregular engagement with online health resources limits their ability to stay informed about health issues. This sporadic usage can be driven by several factors, including low digital literacy, time constraints, and competing priorities. Studies have demonstrated that frequent internet use is positively associated with improved health outcomes. Regular use of the Internet for health information supports better health decision-making, self-management, and health monitoring, emphasizing the need for more consistent engagement with online health resources to optimize health benefits [22,28,40].

Other significant barriers to using the Internet for health information include the lack of E-health literacy and computer literacy. Proficiency in E-health or computer literacy is a key determining factor in utilizing the Internet for health information. This suggests that respondents who are literate in E-health are enthusiastic about searching for health-related information online. In this scenario, those respondents who were E-health literate demonstrated a higher likelihood of using the internet for health information. This observation underscores the importance of digital literacy in facilitating effective utilization of the internet for accessing health-related content [25,28,45].

The field of study plays a critical role in shaping internet use for health information among students. Evidence suggests that health science students are more likely to utilize the internet for health-related information compared to their peers in non-health disciplines. This higher association can be attributed to their academic and professional focus on health, which fosters heightened awareness and a greater need for accessing health information online.

These findings underscore the importance of promoting awareness among students in other disciplines about the benefits and relevance of using the Internet for reliable health information. Awareness campaigns tailored to non-health students can help bridge the gap by encouraging them to leverage online health resources for personal well-being and academic purposes. Such initiatives could enhance health literacy across diverse academic fields, supporting a broader understanding of the importance of credible online health information [27].

Furthermore, the lack of information search skills emerged as a barrier to using the internet for health information seeking. The study participants encompassed individuals from diverse backgrounds, ranging from social science to health science students, including those in technology-related fields. This variation in participant backgrounds may contribute to differences in information search skills, impacting their ability to effectively use the internet for seeking health information. Recognizing and addressing this barrier becomes essential in ensuring that individuals across various academic disciplines can navigate and utilize the internet proficiently for health-related information [22,37,38].

## 5. Strengths and limitations of the study

This study systematically reviews the available literature on the use of the Internet for health information across African countries, representing a pioneering effort to consolidate evidence on this topic from the continent. A key strength of this review is its comprehensive approach, which identifies common barriers to internet utilization for health information, such as limited access, low digital literacy, and challenges in assessing the credibility of online sources. By synthesizing findings from diverse African contexts, the study provides valuable insights that can inform policies and interventions tailored to the region's unique challenges and opportunities. Additionally, adherence to the PRISMA guidelines ensures the methodological rigor and transparency of the review process.

Despite its contributions, the study has certain limitations. The small number of included studies may restrict the generalizability of the findings, and the reliance on cross-sectional studies limits the ability to infer causality between the identified barriers and internet usage patterns. Nonetheless, the study highlights critical gaps in the literature and underscores the need for more extensive and diverse research in this area. Future studies incorporating longitudinal designs and a broader range of countries will help validate and expand upon these findings, providing a stronger foundation for evidence-based interventions.

## 6. Conclusion

In this paper, barriers to using the Internet for health information in African countries were identified through a systematic review of the literature. Lack of smart electronic device ownership, Low frequent use of the internet, Lack of internet access, Lack of E-health literacy, Lack of computer literacy, slow internet connection, Cost of internet access, Lack of information search skills, and rural residence were the most barriers to use the internet for health information.

### Recommendation

Enhance Internet Access and Affordability: Focus on improving internet infrastructure and connectivity, especially in rural areas, to ensure more reliable and faster internet. Additionally, reduce the cost of internet services by introducing subsidies or affordable data plans, making internet access more accessible for everyone.

Increase Smart Device Ownership: Promote initiatives to make smart electronic devices more affordable and widely available. Collaborate with device manufacturers and mobile operators to offer subsidized or donated devices to low-income individuals.

Improve Digital Literacy: Launch educational programs to boost e-health and computer literacy. Provide practical training on using digital health resources and developing effective information search skills, tailored to different literacy levels.

Support Local Content Development: Encourage the creation and dissemination of health information in local languages and culturally relevant formats to enhance the accessibility and relevance of online health resources.

Strengthen Policy and Regulatory Frameworks: Develop and implement policies that support the growth of digital health services and ensure equitable access to online health information.

### Acknowledgments

We extend special thanks to all authors of the studies included in this systematic review and individual articles.

## Author contributions

**Conceptualization:** Alex Ayenew Chereka, Adamu Ambachew Shibabaw, Fikadu Wake Butta, Mathias Nega Tadesse, Mekashaw Tareke Abebe, Fekadu Ayelgn Atanie, Gemeda Wakgari Kitil.

**Data curation:** Alex Ayenew Chereka, Adamu Ambachew Shibabaw, Mathias Nega Tadesse, Mekashaw Tareke Abebe, Fekadu Ayelgn Atanie, Gemeda Wakgari Kitil.

**Formal analysis:** Alex Ayenew Chereka, Adamu Ambachew Shibabaw, Fikadu Wake Butta, Mathias Nega Tadesse, Fekadu Ayelgn Atanie, Gemeda Wakgari Kitil.

**Funding acquisition:** Alex Ayenew Chereka.

**Investigation:** Alex Ayenew Chereka.

**Methodology:** Alex Ayenew Chereka.

**Project administration:** Alex Ayenew Chereka.

**Resources:** Alex Ayenew Chereka, Gemeda Wakgari Kitil.

**Software:** Alex Ayenew Chereka, Fikadu Wake Butta, Mathias Nega Tadesse, Gemeda Wakgari Kitil.

**Supervision:** Alex Ayenew Chereka, Mekashaw Tareke Abebe, Fekadu Ayelgn Atanie, Gemeda Wakgari Kitil.

**Validation:** Alex Ayenew Chereka, Adamu Ambachew Shibabaw, Fikadu Wake Butta, Mathias Nega Tadesse, Mekashaw Tareke Abebe, Gemeda Wakgari Kitil.

**Visualization:** Alex Ayenew Chereka, Adamu Ambachew Shibabaw, Mathias Nega Tadesse, Mekashaw Tareke Abebe, Fekadu Ayelgn Atanie, Gemeda Wakgari Kitil.

**Writing – original draft:** Alex Ayenew Chereka, Fikadu Wake Butta, Mekashaw Tareke Abebe, Fekadu Ayelgn Atanie, Gemeda Wakgari Kitil.

**Writing – review & editing:** Alex Ayenew Chereka, Adamu Ambachew Shibabaw, Fikadu Wake Butta, Mathias Nega Tadesse, Mekashaw Tareke Abebe, Fekadu Ayelgn Atanie, Gemeda Wakgari Kitil.

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
