## [Decision Letter · Decision Letter 0]

14 Aug 2024

PDIG-D-24-00233

Explore barriers to using the internet for health information access in African countries: A Systematic Review.

PLOS Digital Health

Dear Alex Ayenew Chereka,

Thank you for submitting your manuscript to PLOS Digital Health. After careful consideration, we feel that it has merit but does not fully meet PLOS Digital Health's publication criteria as it currently stands. Therefore, we invite you to submit a revised version of the manuscript that addresses the points raised during the review process.

Please submit your revised manuscript within 30 days. If you will need more time than this to complete your revisions, please reply to this message or contact the journal office at digitalhealth@plos.org. Please include the following items when submitting your revised manuscript:

We look forward to receiving your revised manuscript.

Kind regards,

Ana Luísa Neves

Academic Editor

PLOS Digital Health

Journal Requirements:

1. In the online submission form, you indicated that "All the data used for the current study are available from the corresponding author upon reasonable request.". 

a. In a public repository, 

b. Within the manuscript itself, or 

c. Uploaded as supplementary information.

Additional Editor Comments (if provided):

Reviewers' comments:

Reviewer's Responses to Questions

**Comments to the Author**

1. Does this manuscript meet PLOS Digital Health’s publication criteria ? Is the manuscript technically sound, and do the data support the conclusions? The manuscript must describe methodologically and ethically rigorous research with conclusions that are appropriately drawn based on the data presented.

Reviewer #1: Yes

Reviewer #2: Partly

Reviewer #3: Yes

2. Has the statistical analysis been performed appropriately and rigorously?

Reviewer #1: Yes

Reviewer #2: N/A

Reviewer #3: Yes

3. Have the authors made all data underlying the findings in their manuscript fully available (please refer to the Data Availability Statement at the start of the manuscript PDF file)?

Reviewer #1: Yes

Reviewer #2: Yes

Reviewer #3: Yes

4. Is the manuscript presented in an intelligible fashion and written in standard English?

Reviewer #1: Yes

Reviewer #2: Yes

Reviewer #3: Yes

5. Review Comments to the Author

Reviewer #1: Dear Editor and author,

I would like to extend my heartfelt thanks to Ana Luisa Neves, the academic editor of PLOS Digital Health, for inviting me to review the manuscript titled "Explore barriers to using the internet for health information access in African countries: A Systematic Review" for PLOS Digital Health. It was an honor to be entrusted with this task.

Reviewing this manuscript was an enriching experience. The research fills a crucial gap in the literature and addresses a significant public health issue in Ethiopia. I commend the authors for their meticulous work, and I also commend the editorial team for recognizing its importance.

Thank you once again for this opportunity to contribute to the advancement of knowledge in this field. I appreciate the chance to engage with this important topic. However, I have identified some limitations within the study that need to be addressed before acceptance for publication. Please carefully review the comments I have provided below:

1. Modify Figure 1: Ensure it accurately reflects the data and improves clarity.

2. Editorial Issues: Replace "show figure" and "show Table" with (Fig. 1), (Fig. 2), or (Table 1) for consistency and professionalism.

3. Results Section: Remove the search strategy details, as they are already covered in the Methods section, to avoid redundancy.

4. Citations: Use “Ibegbulam, et al.” rather than “Ibegbulam, I.J., et al.” for better readability and standardization.

5. Patient and Public Involvement: Move this section to the Methods to maintain logical structure.

6. Avoid Abbreviations: Refrain from using abbreviations in captions, titles, or abstracts to ensure clarity for all readers.

7. Recommendations: Clearly separate recommendations from the conclusion to highlight actionable insights.

8. Number Headings Chronologically: Ensure all Heading 1 sections are numbered in chronological order for better organization.

NB: These changes will enhance the manuscript’s clarity, readability, and overall quality, making it a valuable contribution to the field. Thank you for considering my suggestions

Reviewer #2: This study aims to systematically review the literature to explore barriers to accessing health information on the Internet in African countries. The focus of the study is appropriate, in that it seeks to understand the barriers to the use of the internet especially in this ICT era, where an increasing number of the general public including professionals seek health information from the internet. The use of PRISMA to guide the methodology is commendable; the inclusion and exclusion criteria where clearly outlined and the selection process where transparent with multiple reviewers, which minimizes bias.

However, there are some fundamental gaps in the analysis and conclusion of the study. The inclusion of articles published in 2006 and even the two published in 2017 and 2018 in a review done and to be published in 2024 on a fast-paced issue such as the topic of this study, limits the significance and relevance of conclusions of the study. The analysis should also have been more in-depth than what was presented. The respondents in the articles reviewed were either students (three articles) or health workers (two articles) and this should have been factored in the analysis since the challenges the two groups faced might have been different and could have influenced the conclusions. Moreso there were no articles that respondents were from other categories of the general public and this this should have been factored in the analysis, discussions and conclusions. The analysis is thus rather too simplistic - it just describes the number of journals that mentioned the different barriers and then made policy recommendations based on that.

I would thus suggest that the authors explore further refining their search criteria to possibly get more articles. The analysis and discussion should also be more in-depth to justify the conclusions and recommendations made.

Reviewer #3: Please, I don't any review comments to the Author/s. All my comments have been addressed in the review comments I have attached. Hence, I have no additional information aside the comments I have attached.

6. PLOS authors have the option to publish the peer review history of their article (what does this mean? ). If published, this will include your full peer review and any attached files.

**Do you want your identity to be public for this peer review?** For information about this choice, including consent withdrawal, please see our Privacy Policy .

Reviewer #1: No

Reviewer #2: No

Reviewer #3: Yes: Alexander Laar

While revising your submission, please upload your figure files to the Preflight Analysis and Conversion Engine (PACE) digital diagnostic tool, https://pacev2.apexcovantage.com/ . PACE helps ensure that figures meet PLOS requirements. To use PACE, you must first register as a user. Registration is free. Then, login and navigate to the UPLOAD tab, where you will find detailed instructions on how to use the tool. If you encounter any issues or have any questions when using PACE, please email PLOS at figures@plos.org. Please note that Supporting Information files do not need this step.

---

## [Decision Letter · Decision Letter 1]

6 Nov 2024

PDIG-D-24-00233R1Explore barriers to using the internet for health information access in African countries: A Systematic Review.PLOS Digital Health Dear Dr. Chereka, Thank you for submitting your manuscript to PLOS Digital Health. After careful consideration, we feel that it has merit but does not fully meet PLOS Digital Health's publication criteria as it currently stands. Therefore, we invite you to submit a revised version of the manuscript that addresses the points raised during the review process. Please submit your revised manuscript within 30 days Dec 06 2024 11:59PM. If you will need more time than this to complete your revisions, please reply to this message or contact the journal office at digitalhealth@plos.org. Please include the following items when submitting your revised manuscript:* A rebuttal letter that responds to each point raised by the editor and reviewer(s). You should upload this letter as a separate file labeled 'Response to Reviewers '. This file does not need to include responses to any formatting updates and technical items listed in the 'Journal Requirements' section below.* A marked-up copy of your manuscript that highlights changes made to the original version. You should upload this as a separate file labeled 'Revised Manuscript with Track Changes '.* An unmarked version of your revised paper without tracked changes. You should upload this as a separate file labeled 'Manuscript '. If you would like to make changes to your financial disclosure, competing interests statement, or data availability statement, please make these updates within the submission form at the time of resubmission. Guidelines for resubmitting your figure files are available below the reviewer comments at the end of this letter. We look forward to receiving your revised manuscript. Kind regards, Laura Sbaffi, PhD, MA, MScSection EditorPLOS Digital Health Leo Anthony CeliEditor-in-ChiefPLOS Digital Healthorcid.org/0000-0001-6712-6626  **Journal Requirements:** **Additional Editor Comments (if provided):****Reviewers' Comments:** Reviewer's Responses to Questions

**Comments to the Author**

1. If the authors have adequately addressed your comments raised in a previous round of review and you feel that this manuscript is now acceptable for publication, you may indicate that here to bypass the “Comments to the Author” section, enter your conflict of interest statement in the “Confidential to Editor” section, and submit your "Accept" recommendation.

Reviewer #1: All comments have been addressed

Reviewer #2: (No Response)

Reviewer #3: All comments have been addressed

2. Does this manuscript meet PLOS Digital Health’s publication criteria ? Is the manuscript technically sound, and do the data support the conclusions? The manuscript must describe methodologically and ethically rigorous research with conclusions that are appropriately drawn based on the data presented.

Reviewer #1: Yes

Reviewer #2: Partly

Reviewer #3: Yes

3. Has the statistical analysis been performed appropriately and rigorously?

Reviewer #1: Yes

Reviewer #2: No

Reviewer #3: Yes

4. Have the authors made all data underlying the findings in their manuscript fully available (please refer to the Data Availability Statement at the start of the manuscript PDF file)?

Reviewer #1: Yes

Reviewer #2: Yes

Reviewer #3: Yes

5. Is the manuscript presented in an intelligible fashion and written in standard English?

Reviewer #1: Yes

Reviewer #2: Yes

Reviewer #3: Yes

6. Review Comments to the Author

Reviewer #1: (No Response)

Reviewer #2: I would like to appreciate the authors for the detailed response to my initial feedback. I particularly acknowledge their response as to why the older articles should and were included in the review. Also, their effort to widen the scope of the search to include more articles and the limitation to doing this are also acknowledged.

However, i believe the analysis of the findings and discussion are still not robust enough even based on the fewer number of articles reviewed. There are also inconsistencies in their feedback and inferences. some of my concerns are listed below

1. While Table one provided the characteristics of the 6 articles that met the inclusion criteria, Table 2 which describes the frequency of the barriers mentioned in the articles, made mention of 8 articles with reference numbers 20, 22, 25, 27, 28, 37, 38 and 39. Even though 20 and 39 are the same articles; 27 in a distinct article, thus making the articles 7. This is a red flag for me.

2. The authors also mention that they “revisited our analysis with your comments in mind to explore potential differences between the challenges faced by students and health workers. While we agree that these groups might encounter different challenges, our analysis did not reveal significant differences in the context of the studies reviewed”. But looking at the data presented in Table 2, the study with reference 38 which was on secondary school students in Nigeria reported 8 out of the 11 barriers while college students in Ghana (37) reported 10 out of the 11 barriers while the study on (20 and 39) reported three barriers. This sorts of show clear distinction in the magnitude and even types of barriers experienced by the different categories, and I expect the authors to have looked more into these findings and provide possible insights into factors that might have been responsible for the findings e.g. policy environment, socio-economic factors, Location etc

3. Also, in their response to the need to widen the scope of the search, the authors “observed that while the titles of the newly identified articles were similar to those already included, their results and contents differed significantly. Due to these differences, we have opted to retain the original set of articles in order to maintain consistency and relevance to our research objectives”. This sort of implies some form of bias in the application of the inclusion criteria.

4. I am also curious about the barrier “low frequent use of the internet” as a distinct barrier. This sound like a circular reference since almost all the other barriers could lead to infrequent use of the internet.

There are other observations in the discussions and even in the background section. I thus believe that more works needs to be done on the article to meet the rigour expected of an article to be published in PLOS.

Reviewer #3: (No Response)

7. PLOS authors have the option to publish the peer review history of their article (what does this mean? ). If published, this will include your full peer review and any attached files.

**Do you want your identity to be public for this peer review?** For information about this choice, including consent withdrawal, please see our Privacy Policy .

Reviewer #1: No

Reviewer #2: No

Reviewer #3: **Yes: ** Alexander Suuk Laar

 **Figure resubmission:** While revising your submission, please upload your figure files to the Preflight Analysis and Conversion Engine (PACE) digital diagnostic tool, https://pacev2.apexcovantage.com/ . PACE helps ensure that figures meet PLOS requirements. To use PACE, you must first register as a user. Registration is free. Then, login and navigate to the UPLOAD tab, where you will find detailed instructions on how to use the tool. If you encounter any issues or have any questions when using PACE, please email PLOS at figures@plos.org. Please note that Supporting Information files do not need this step. If there are other versions of figure files still present in your submission file inventory at resubmission, please replace them with the PACE-processed versions. **Reproducibility:** To enhance the reproducibility of your results, we recommend that authors of applicable studies deposit laboratory protocols in protocols.io, where a protocol can be assigned its own identifier (DOI) such that it can be cited independently in the future. Additionally, PLOS ONE offers an option to publish peer-reviewed clinical study protocols. Read more information on sharing protocols at https://plos.org/protocols?utm_medium=editorial-email&utm_source=authorletters&utm_campaign=protocols

---

## [Editor Report · Decision Letter 2]

19 Nov 2024

PDIG-D-24-00233R2Explore barriers to using the internet for health information access in African countries: A Systematic Review.PLOS Digital Health Dear Dr. Chereka, Thank you for submitting your manuscript to PLOS Digital Health. After careful consideration, we feel that it has merit but does not fully meet PLOS Digital Health's publication criteria as it currently stands. Therefore, we invite you to submit a revised version of the manuscript that addresses the points raised during the review process. Please submit your revised manuscript within 30 days Dec 19 2024 11:59PM. If you will need more time than this to complete your revisions, please reply to this message or contact the journal office at digitalhealth@plos.org. Please include the following items when submitting your revised manuscript:* A rebuttal letter that responds to each point raised by the editor and reviewer(s). You should upload this letter as a separate file labeled 'Response to Reviewers '. This file does not need to include responses to any formatting updates and technical items listed in the 'Journal Requirements' section below.* A marked-up copy of your manuscript that highlights changes made to the original version. You should upload this as a separate file labeled 'Revised Manuscript with Track Changes '.* An unmarked version of your revised paper without tracked changes. You should upload this as a separate file labeled 'Manuscript '. If you would like to make changes to your financial disclosure, competing interests statement, or data availability statement, please make these updates within the submission form at the time of resubmission. Guidelines for resubmitting your figure files are available below the reviewer comments at the end of this letter. We look forward to receiving your revised manuscript. Kind regards, Laura Sbaffi, PhD, MA, MScSection EditorPLOS Digital Health Leo Anthony CeliEditor-in-ChiefPLOS Digital Healthorcid.org/0000-0001-6712-6626  **Journal Requirements:** **Additional Editor Comments (if provided):****Reviewers' Comments:**   **Figure resubmission:** While revising your submission, please upload your figure files to the Preflight Analysis and Conversion Engine (PACE) digital diagnostic tool, https://pacev2.apexcovantage.com/ . PACE helps ensure that figures meet PLOS requirements. To use PACE, you must first register as a user. Registration is free. Then, login and navigate to the UPLOAD tab, where you will find detailed instructions on how to use the tool. If you encounter any issues or have any questions when using PACE, please email PLOS at figures@plos.org. Please note that Supporting Information files do not need this step. If there are other versions of figure files still present in your submission file inventory at resubmission, please replace them with the PACE-processed versions. **Reproducibility:** To enhance the reproducibility of your results, we recommend that authors of applicable studies deposit laboratory protocols in protocols.io, where a protocol can be assigned its own identifier (DOI) such that it can be cited independently in the future. Additionally, PLOS ONE offers an option to publish peer-reviewed clinical study protocols. Read more information on sharing protocols at https://plos.org/protocols?utm_medium=editorial-email&utm_source=authorletters&utm_campaign=protocols

---

## [Editor Report · Decision Letter 3]

9 Dec 2024

Explore barriers to using the internet for health information access in African countries: A Systematic Review.

PDIG-D-24-00233R3

Dear Mr Chereka,

We are pleased to inform you that your manuscript 'Explore barriers to using the internet for health information access in African countries: A Systematic Review.' has been provisionally accepted for publication in PLOS Digital Health.

Best regards,

Laura Sbaffi, PhD, MA, MSc

Section Editor

PLOS Digital Health